# Effects of Organic Acids, Amino Acids and Phenolic Compounds on Antioxidant Characteristic of Zhenjiang Aromatic Vinegar

**DOI:** 10.3390/molecules24203799

**Published:** 2019-10-22

**Authors:** Bo Zhang, Ting Xia, Wenhui Duan, Zhujun Zhang, Yu Li, Bin Fang, Menglei Xia, Min Wang

**Affiliations:** State Key Laboratory of Food Nutrition and Safety, Key Laboratory of Industrial Fermentation Microbiology, College of Biotechnology, Tianjin University of Science and Technology, Tianjin 300222, China; bozhango@163.com (B.Z.); xiatingsyu@foxmail.com (T.X.); duanwhyy@163.com (W.D.); yzhujunz@126.com (Z.Z.); liyu962366@163.com (Y.L.); ab1772850431@163.com (B.F.); mlxia@tust.edu.cn (M.X.)

**Keywords:** Zhenjiang aromatic vinegar, organic acids, amino acids, phenolic compounds, antioxidant activity, synergistic effect

## Abstract

Zhenjiang aromatic vinegar (ZAV) is one of the famous Chinese vinegars, which contains various physicochemical and bioactive compositions. In the present study, physicochemical properties and total antioxidant activity were detected in ZAV samples. The correlation between of organic acids, amino acids, phenolic compounds, and the antioxidant activity of ZAV were explored. The results showed that contents of total acids, soluble solids, reducing sugar and total antioxidant activity in ZAV were increased with aging time, and those in ZAV-5 were the highest. Organic acids and amino acids exhibited weak antioxidant activity, while phenolic compounds had higher antioxidant ability. In addition, amino acids had synergistic effect on the antioxidant activity of phenolic compounds, whereas organic acids inhibited the antioxidant activity of phenolic compounds. Moreover, it was found that phenolic compounds including catechin, vanillic acid and syringic acid showed higher contribution rates to antioxidant activities of mixed phenolic compounds. In conclusion, these findings would provide references to control the antioxidant characteristic of vinegar through regulating the main compositions, and further improve the quality of vinegar production.

## 1. Introduction

Vinegar has been used as acidic condiment all over the world for more than three thousand years [1]. Traditional vinegars were brewed from different raw materials such as fruits and grains by spontaneous fermentation technology [2,3]. Zhenjiang aromatic vinegar (ZAV), one of the most famous vinegars in China, is produced by solid-state fermentation [4]. Raw materials of ZAV include rice, wheat bran and rice hulls. The brewing process of ZAV is mainly divided into five stages: alcoholic fermentation (AF), acetic acid fermentation (AAF), leaching, decoction and aging stage [5]. During the process, it produces many chemical compositions such as organic acids, amino acids, sugars and other bioactive compounds, which have benefit effects on human health [6,7,8,9].

Vinegars contain abundant organic acids, which are the main source of vinegar flavor and important indictors to evaluate the quality of vinegar [10]. They can be divided into volatile and non-volatile acids. Volatile organic acids include acetic acid, formic acid, propionic acid, butyric acid and quinic acid, etc., [11,12]. It has been reported that acetic acid is the most abundant organic acid in vinegar, accounting for about half of the total organic acids [13]. In addition, there are a large number of non-volatile acids such as lactic acid, malic acid, pyroglutamic acid, citric acid and succinic acid [14,15]. These non-volatile acids buffer the stimulation of acetic acid and contribute to the strong aroma and flavor [13]. 

Amino acids in vinegars including free amino acids and nonprotein amino acids, come from raw materials and microbial decomposition during the fermentation and aging processes [16,17,18]. Amino acids can provide different taste to vinegars such as umami, sweet and bitter [19]. Amino acids are essential nutrients, which play important roles to human health [20]. Some amino acids including lysine, histidine and arginine react with sugars and produce melanoidins, which have antioxidant activity [17]. Otherwise, sulfur-containing amino acids such as methionine and cysteine can be converted into reduced glutathione through redox cycle, which subsequently protect against oxidative damage [21]. Aromatic amino acids including tryptophan, tyrosine and phenylalanine inhibit chain reaction of free radical and have antioxidant effect [22].

Phenolic compounds are major bioactive ingredients in vinegars, which mainly derived from raw materials [23]. These phenolic compounds contain one or more phenolic hydroxyl groups on the aromatic ring and play an important role in the antioxidant activity [24,25]. Recent studies have reported that the antioxidant activity of vinegars has positive correlation with polyphenolic fraction [26,27,28]. Bakir et al. [29] reported that antioxidant activities of grape and apple vinegars range from 254 ± 8 mg TEAC/100 mL to 568 ± 76 mg TEAC/100 mL by ferric reducing antioxidant power (FRAP) assay and from 418 ± 49 mg TEAC/100 mL to 2561 ± 260 mg TEAC/100 mL by 2,2′-azino-bis(3-ethylbenzthi azoline-6-sulfonic acid) (ABTS) assay, respectively. The phenolic compounds in these vinegars were detected including gallic acid, *p*-hydroxybenzoic acid, catechin, syringic acid, caffeic acid and *p*-coumaric acid. Verzelloni et al. [30] found that the antioxidant capacity of red wines and traditional balsamic vinegars was highly correlated with their phenolic and flavonoid contents. The compounds contained catechin, epigallocatechin galate (EGCG), quercetin, cinnamic acid, gallic acid and resveratrol. In a previous study, we found that the antioxidant capacity of Shanxi aged vinegar was highly correlated with their polyphenol contents. There were eight phenolic compounds in Shanxi aged vinegar determined by high performance liquid chromatography (HPLC) [31]. However, the major compositions and their contribution to antioxidant activities of ZAV have been rarely explored.

The aim of this study is to elucidate the antioxidant properties of ZAV and their potentially correlated with the main chemical compositions. In this study, the contents of physicochemical components, organic acids, amino acids and polyphenols were detected in ZAV. In addition, the antioxidant activities of ZAV and major compounds including organic acids, amino acids and polyphenols were evaluated by different methods. Moreover, the relationship between these main chemical compositions and the antioxidant properties of ZAV was investigated. 

## 2. Results and Discussion

### 2.1. Physicochemical Parameters in ZAV

Physicochemical parameters of ZAV were shown in Table 1. pH values in all vinegar samples ranged from 3.28 ± 0.02 to 3.46 ± 0.01. The total acids contents ranged between 4.954 ± 0.221 and 7.726 ± 0.017 g/100 mL, which were increased with the aging time. Kim et al. [32] reported that pH and total acids contents of commercial vinegar in Korea were 2.81~3.20 and 2.07 ± 0.03~2.42 ± 0.02 g/100 mL, respectively. Xia et al. [27] reported that the total acids content was 3.5 ~ 8.0 g/100 mL in Shanxi aged vinegar. By comparing the data of total acids contents in different types of vinegars, total acids content of ZAV is higher than that in Korean vinegar, and similar to that in Shanxi aged vinegar. In addition, contents of non-volatile acids in ZAV ranged from 1.769 ± 0.057 to 2.388 ± 0.089 g/100 mL, which were increased during six years of aging time and then decreased gradually. Non-volatile acids and reducing sugar can buffer the stimulation of acetic acid and improve the taste and flavor of vinegar [13]. Reducing sugar contents ranged between 1.450 ± 0.071 and 2.850 ± 0.212 g/100 mL, which were positively associated with aging time. The contents of soluble solids in the samples were 12.738 ± 0.084 ~ 19.428 ± 0.005 g/100 mL, and elevated with the aging time. Amino nitrogen is content of nitrogen existed as amino acids, which is a characteristic index to determine the fermented degree in the products [17]. The results showed that content of amino nitrogen in the samples ranged between 0.399 ± 0.002 and 0.468 ± 0.021 g/100mL, which decreased slightly during three years of aging time and then increased with the aging time. Collectively, the contents of total acids, soluble solids and reducing sugar were increased with aging time, which were the highest in ZAV-5.

### 2.2. Analysis of Total Antioxidant Activity in ZAV

Total antioxidant activities of ZAV in different samples were measured by DPPH, ABTS and FRAP methods. As shown in Figure 1, the values of antioxidant activity ZAV were 16.876 ± 1.384 ~ 33.172 ± 3.703 mmol TEAC/L (2,2-diphenyl-1-picrylhydrazyl (DPPH) assay), 24.032 ± 2.650 ~ 35.608 ± 2.481 mmol TEAC/L (ABTS assay) and 26.635 ± 1.654 ~ 38.844 ± 2.282 mmol TEAC/L (FRAP assay), respectively. These values were followed in decreasing order: ZAV-5 > ZAV-4 > ZAV-3 > ZAV-2 > ZAV-1. Bertelli et al. [33] reported that the antioxidant activities of traditional balsamic vinegar from Modena (>25 years of aging) were 19.61 ± 14.0 μM TEs/mL by DPPH and 33.04 ± 21.8 μM TEs/mL by ABTS. The antioxidant activities of Shanxi aged vinegar in different aging time ranged from 8.13 ± 0.29 mmol TEAC/L to 22.11 ± 1.53 mmol TEAC/L by FRAP and 6.10 ± 0.62 mmol TEAC/L to 29.94 ± 0.58 mmol of TEAC/L by ABTS, respectively, which were increased with the aging time [31]. In this study, ZAV showed similar trends compared with previous studies, and total antioxidant activity of ZAV-5 was the highest in all vinegar samples. 

### 2.3. Organic Acids Contents and Antioxidant Activities in ZAV-5

Then, we focused on the principal component in ZAV-5. Vinegars contain abundant organic acids, which mainly come from brewing raw materials [17]. Organic acids in ZAV-5 were presented in Table 2. It was found that 7 organic acids in ZAV-5 were identified by HPLC. Among these organic acids, the content of acetic acid was 6.606 ± 0.352 g/100 mL, was the highest in ZAV-5. In addition, the contents of lactic acid and pyroglutamic acid were 3.283 ± 0.241 and 1.363 ± 0.255 g/100 mL, respectively, indicating the major non-volatile acids in ZAV-5. Meanwhile, total contents of non-volatile acids in ZAV-5 accounted for about 85.85% of acetic acid content. Zhu et al. [10] reported that 10 kinds of organic acids were identified in Shanxi aged vinegar by ion liquid chromatography (IC). Among these organic acids, the content of acetic acid was the highest in volatile acids, which ranged from 45945.9 mg/L to 87723.5 mg/L. Lactic acid content was the largest in nonvolatile acid, which varied between 2575.7 mg/L and 30336.6 mg/L. Lalou et al. [34] found that acetic acid, formic acid, citric acid, malic acid and succinic acid in balsamic vinegar were identified by nuclear magnetic resonance (NMR) method. The amount of acetic acid was 28.52 ± 2.14 ~ 63.57 ± 5.80 g/kg, followed by malic acid (1.34 ± 0.054 ~ 8.59 ± 0.438 g/kg) and citric acid (0.29 ± 0.008 ~ 2.90 ± 0.123 g/kg). By comparison of organic acids in different types of vinegars, acetic acid as a kind of volatile acid is the major organic acid in vinegars, which contributes the primary flavor of vinegars. The kinds and contents of non-volatile acids are different in various vinegars, which may associate with different raw materials and fermentation technologies during manufacturing process.

To explore the relationship between the main ingredients and antioxidant activities in ZAV-5, the antioxidant activity values of organic acids were evaluated by DPPH, ABTS and FRAP. These results showed that all organic acids exhibited weak iron reducing capacity, but were not detected in ABTS assay. Acetic acid and oxalic acid showed weak DPPH scavenging ability, and the rest organic acids were not detected in DPPH assay (Table 2). These results indicate that organic acids exhibit very low contribution to total antioxidant activity of ZAV-5.

### 2.4. Amino Acids Contents and Antioxidant Activities in ZAV-5

As shown in Table 3, amino acids in ZAV-5 were detected by amino acid analyzer. The content of leucine (1.535 ± 0.028 g/L) was the highest among all amino acids, followed by glycine (1.444 ± 0.221 g/L) and threonine (1.411 ± 0.324 g/L), respectively. These amino acids were mainly contributed to sweet and fresh taste of ZAV. Additionally, essential amino acids accounted for about 61.52% of total amino acids in ZAV. These amino acids are essential nutrients to human body, which have beneficial effects such as improving immunity and promoting brain development [7,20]. It has been reported that the content of alanine in Shanxi aged vinegar was in the range of 1328.96 ± 6.13 ~ 2990.49 ± 42.56 µg/mL, which was the highest amino acids. Leucine (699.23 ± 6.57~1832.29 ± 11.82 µg/mL) and glutamic acid (92.55 ± 3.59~1635.48 ± 29.26 µg/mL) were the other two major amino acids [27]. In a study conducted by Chinnici et al. [35] who studied amino acids in traditional balsamic vinegar (TBV), balsamic vinegar and sherry vinegar. It was found that proline was the highest content (494.67 mg/kg, 98.8 mg/kg and 308.56 mg/kg) in three types of vinegars. According to previous studies, the contents of amino acids in various vinegars were different related to raw materials and production process. Among these amino acids, leucine, glycine and threonine were the main amino acids in ZAV-5.

Then, the antioxidant activities of amino acids in ZAV-5 were detected by DPPH, ABTS and FRAP. The results showed that phenylalanine, tyrosine and lysine had low DPPH and ABTS radicals scavenging ability and iron reducing ability. Arginine had weak scavenging ability and iron reducing ability. However, the rest amino acids had very faint iron reducing ability, and were not examined by DPPH and ABTS. The data imply that amino acids contribute weakly to total antioxidant activity of ZAV-5.

### 2.5. Phenolic Compounds Contents and Antioxidant Activities in ZAV-5

The polyphenols in ZAV-5 were determined by HPLC (Table 4). The results showed that 11 polyphenols were detected in ZAV-5. Among these polyphenols, the content of catechin (36.745 ± 1.459 mg/L) was highest, followed by *p*-hydroxybenzoic acid (24.738 ± 2.376 mg/L) and vanillic acid (15.223 ± 3.892 mg/L) in ZAV-5. Previous study reported that gallic acid (114.67 ± 3.88 mg/L), catechins (86.98 ± 0.36 mg/L) and caffeic acid (53.69 ± 1.87 mg/L) were the main phenolic compounds in Shanxi aged vinegar [31]. Kharchoufi et al. [36] reported that there were 17 phenolic compounds in pomegranate vinegar determined by ultra-performance liquid chromatography-mass spectrum (UPLC-MS). The content of protocatechuic acid (28.88 ± 0.02 mg/L) was highest, followed by gallic acid. In this study, the content of gallic acid was 0.981 ± 0.092 mg/L, which were not different from other studies. Taken together, these results indicate that catechin, *p*-hydroxybenzoic acid and vanillic acid were the major phenolic compounds in ZAV-5. 

The antioxidant activities of phenolic compounds were showed in Table 4. It was found that the values of antioxidant activities in DPPH method were followed by caffeic acid, gallic acid, catechin, sinapic acid, syringic acid, ferulic acid and rutin, which had high antioxidant activity. According to ABTS assay, gallic acid had the highest antioxidant activity, which was not directly proportional to the content, followed by catechin, rutin, ferulic acid, syringic acid, sinapic acid and caffeic acid. The antioxidant activity values of rutin and gallic acid were higher than others, and as follows in decreasing order: rutin > gallic acid > syringic acid > caffeic acid > ferulic acid > sinapic acid > catechin. However, vanillic acid, *p*-coumaric acid and *p*-hydroxybenzoic acid had very low antioxidant activity (FRAP assay). Taken together, the results suggest that polyphenols including caffeic acid, gallic acid, sinapic acid, syringic acid, catechin, rutin and ferulic acid exhibit high antioxidant activity in ZAV-5.

### 2.6. Effects of Main Components on Antioxidant Activities of ZAV-5

To investigate the effects of the main components on the antioxidant activities in ZAV-5, organic acids, amino acids and phenolic compounds were mixed and determined by DPPH, ABTS and FRAP. As shown in Table 5, the antioxidant activities of organic acid solution were very low in FRAP assay, but not be detected in DPPH and ABTS assay. The antioxidant activity of phenolic compounds was significant higher than amino acids and organic acids. When organic acids and phenolic compounds were mixed, antioxidant activities of the mixture were lower than antioxidant activities of phenolic compounds alone. The results suggest that organic acids have antagonistic effect on the antioxidant capacity of phenolic compounds. When organic acids, amino acids and phenolic compounds were mixed, the antioxidant activities of the three main compounds were lower than antioxidant activities of amino acids and phenolic compounds mixture, which also demonstrated the antagonism of organic acids. In addition, amino acids solution showed low antioxidant capacity by three methods. The antioxidant activities of amino acids and phenolic compounds were higher than those of phenolic compounds alone, indicating amino acids had synergistic effect on the antioxidant capacity of phenolic compounds. When organic acids, amino acids and phenolic compounds were mixed, the antioxidant activities of the three main compounds were higher than antioxidant activities of organic acids and phenolic compounds mixture, which also demonstrated the synergism of amino acids. In general, these data suggest that phenolic compounds showed the highest antioxidant activity among the three main compounds. Amino acids showed a synergistic effect on the antioxidant activity of phenolic compounds, while organic acids exhibited an antagonistic effect.

### 2.7. Contribution of Phenolic Compounds to Antioxidant Activity in ZAV-5

To further verify the contribution of phenolic compounds on antioxidant capacity in ZAV-5, the antioxidant activities of mixed solution were measured by three different assays. As shown in Figure 2, catechin, vanillic acid and syringic acid were the main phenolic compounds, which had high contribution rates. Catechin and vanillic acid were higher contents among these phenolic compounds and had a high contribution rate. Content of syringic acid was lower, while the contribution was higher. It may be due to the synergistic effect among the phenolic compounds. In addition, the contribution rates of the phenolic compounds in the mixed solution were far more than 100%, which revealed that phenolic compounds reacted with each other had synergistic effect on the antioxidant activity. Chen et al. [37] detected that there were eight free phenolic acids in Shanxi aged vinegar. Among these phenolic compounds, except for salicylic acid or *p*-hydroxybenzoic acid with protocatechuic acid or dihydro ferulic acid showed antagonistic effect, other phenolic compounds reacted with each other and showed synergistic effect. The contribution of phenolic compounds to antioxidant activity in vinegars has been rarely reported. The results imply that contributions of phenolic compounds to antioxidant activity were not only correlated with their contents, but with synergistic effect on other compounds. More studies are needed to clarify the antioxidant components and their contribution rates to antioxidant activity of vinegar.

## 3. Materials and Methods

### 3.1. Chemical Reagents and Vinegar Samples

The FRAP and ABTS assay kits were supplied from Beyotime Institute of Biotechnology (Shanghai, China), which were used to measure antioxidant activity. Standards of organic acids and phenolic compounds were purchased from Sigma-Aldrich (Deisenhofen, Germany). Folin-Ciocalteu reagent and DPPH were purchased from Sinopharm Chemical Reagent Co., Ltd (Shanghai, China). 

6 vinegar samples for each aging time (0, 3, 6, 8 and 10 years) were obtained from Zhenjiang Hengshun Vinegar Co., Ltd (Jiangsu, China). The characteristics of the vinegar samples were shown in Table 6.

### 3.2. Determination of the Physical and Chemical Parameters

The pH values were measured using a pH meter (Metrohm, Herisau, Switzerland). Total acids were determined according to the GB/T 12456-2008. The contents of amino nitrogen in vinegar samples were determined by formaldehyde titration (GB/T 5009.235-2016). The reducing sugar contents in ZAV were calculated by Fehling’s method. The contents of non-volatile organic acids and soluble solids were detected according to the methods of National Standard (GB/T18623-2011; GB/T18187-2000). 

### 3.3. Assessment of Antioxidant Activity

DPPH assay. Every sample was diluted 20 times with distilled water. DPPH working solution was prepared. 20 mg of DPPH was added to 250 mL ethanol to make 0.2 mmol/L DPPH solution. 0.2 mmol/L DPPH solution and ethanol (5:4, *v*/*v*) were mixed to make DPPH working solution. Diluted samples (20 μL) were added to DPPH working solution (180 μL), and the mixed solution were incubated at room temperature for 30 min in the dark. The absorbances of solution were measured at 517 nm using an automated enzyme-linked immunosorbent assay (ELISA) reader (Tecan Austria GmbH, Salzburg, Austria). The data were expressed as trolox equivalent antioxidant capacity (TEAC). 

ABTS assay. The ABTS assay was performed as described by Re et al. [38] with a slight modification. Every sample was diluted 20 times with distilled water. 7 mM ABTS solution was mixed with 2.45 mM potassium persulfate, and then kept at room temperature in the dark for 12 h. ABTS solution was diluted using ethanol to an absorbance of 0.70 ± 0.10 at 414 nm. 170 µL of the diluted ABTS solution was added to 10 µL of an appropriately diluted sample and incubated at room temperature for 6 min. The absorbance was recorded at a wavelength of 414 nm using an ELISA reader. Trolox was calculated as a standard compound. Antioxidant activity was recorded as TEAC. 

FRAP assay. Every sample was diluted 20 times with distilled water. The FRAP solution was prepared by mixing 300 mM acetate buffer, 10 mM tripyridyltriazine (TPTZ) and 20 mM FeCl_3_ with a ratio of 10:1:1 (*v*/*v*/*v*). 180 µL of the FRAP working solution and 5 µL of this diluted sample were mixed, and the mixture was incubated at room temperature for 5 min. The absorbance was measured at 593 nm. Trolox was used as a standard compound. The antioxidant activity was expressed as TEAC.

### 3.4. Measurement of Organic Acids

The organic acid contents of vinegar samples were determined by HPLC (Agilent Corp., Karlsruhe, Germany). The chromatographic column was used an Aminex HPX-87H lon Exculsion Column (7.8 × 300 mm; i.d., 5µm; Shanghai, China). The mobile phase includes H_2_SO_4_-water (5 mM). The injection volume was 20 μL, and the flow rate was 0.6 mL/min with 30 °C column temperature. The ultraviolet (UV) absorption was detected at 215 nm. The identification and detection of each compound were carried by comparison of their retention times with authentic standards. 2 mL of diluted samples were centrifuged at 6000 rpm for 10 min at a room temperature, and the supernatant was filtered through 0.45 µm Millipore membrane (Billerica, MA, USA).

### 3.5. Detection of Amino Acids

Amino acid contents were analyzed by Amino Acid Analyzer (S-433D; SYKAM, Munich, Germany) equipped with a fluorescence detector. Vinegar samples were centrifuged at 7000 rpm for 15 min at a room temperature, diluted 50 times with distilled water. In brief, diluted vinegar samples (200 µL) and salicylic acid (2%, *v*/*v*) (800 µL) were mixed and stood at room temperature for 30 min. The mixture was centrifuged at 13,000 rpm for 15 min, and the upper layer was filtered via 0.45µm Millipore membrane. LCAK06/Na (4.6 mm × 150 mm) column was used for free amino acid determination. The working parameters were 0.45 mL/min flow rate, 50 μL injection volume and 58 °C column temperature. The ninhydrin (NIN)-derivatized amino acids were determinated at 570 nm and at 440 nm. A calibration curve was prepared with different concentrations of amino acids solutions, and the results were expressed as g/L of vinegar samples.

### 3.6. Measurement of Total Phenolic and Flavonoid Contents

Total phenolic contents of vinegar samples with different aging times were determined using the Folin-Ciocalteu method [39]. Briefly, 0.2 mL of an appropriately diluted sample in distilled water (1:10, *v*/*v*) and 0.8 mL of Folin-Ciocalten reagent were mixed and incubated in the dark for 3 min at room temperature. 1.5 mL of 10% sodium carbonate and 7.5 mL of distilled water were mixed and incubated for 2 h. The absorbance was detected at 765 nm. The data were calculated as mg GAE/mL. Gallic acid was used as a standard.

The total flavonoids contents of vinegar samples were measured by the colorimetric assay. Vinegar samples were neutralized by 2% NaOH solution earth firstly, and diluted 10 times with distilled water. 2.0 mL of diluted samples in distilled water and 1.0 mL of NaNO_2_ solution (5%, *w*/*v*) were mixed and kept for 6 min at room temperature in the dark. Then, 1.0 mL of Al(NO_3_)_3_ solution (5%, *w*/*v*) was added and stood for 6 min, and then 4.0 mL of NaOH solution (20%, *w*/*v*) was added. The volume of mixed solution was 25.0 mL distilled with water and kept at room temperature for 15 min. The absorbance was measured by a spectrophotometer at 510 nm. The results were expressed as mg RE/mL (Rutin was used as reference).

### 3.7. Determination of Phenolic Compounds

The phenolic compounds contents were analyzed by HPLC. ZAV were extracted with ethyl acetate three times. Vinegar samples were neutralized with NaOH solution and mixed with ethyl acetate. The ethyl acetate layer was ultrasound and centrifuged for 10 min, respectively. The upper layer was collected and concentrated at 35–40 ℃ using a vacuum rotary evaporator (Labortechnik, AG CH-9230, Postfach, Flawil, Switzerland). Finally, the residue was dissolved in 50% methanol (*v/v*). The supernatant was filtered using a 0.45 μm Millipore membrane. Chromatographic condition: Phenyl chromatography column (250 mm × 4.6 mm i.d., 5 µm). The chromatographic conditions were showed as follows: flow rate was 1.0 mL/min; injection volume was 10 µL, and column temperature was 40 °C. Mobile phase: solvent A: water/acetic acid (98:2, *v*/*v*); solvent B: water/acetonitrile/acetic acid (73:25:2, *v*/*v*/*v*): 0–19 min, 4–5% B; 46–59 min, 15–20% B; 59–61 min, 10–20% B, 64 min, 5% B and then held for 5 min. The UV absorption was detected at 278 nm.

### 3.8. Examination of Antioxidant Activities in Phenolic Mixed Solution

According to the contents of phenolic compounds measured by HPLC, the model solution of phenolic compounds was prepared with standard substances. A kind of phenolic compound was deleted from the model solution, and then antioxidant activities of residual solution were determined by DPPH, ABTS and FRAP methods. The results were calculated as follows: Contribution rate of missing substance (%) = ((Antioxidant value of model solution − antioxidant value of residual solution)/Antioxidant value of model solution) × 100%.

### 3.9. Statistical Analysis

Results were presented as mean ± standard deviation (S.D.). Variance analysis was carried by duncan multiple comparison test using the SPSS statistical package version 24.0 software (SPSS Inc., Chicago, IL, USA). *p* < 0.05 suggested significant differences.

## 4. Conclusions

In the present study, physicochemical properties, organic acids, amino acids, phenolic compounds, and antioxidant characteristic of ZAV were investigated. The results showed that the contents of total acids, soluble solids, reducing sugar and total antioxidant activity in ZAV were increased with aging time, and highest in ZAV-5. In addition, acetic acid, lactic acid and pyroglutamic acid were the main organic acids, and leucine, glycine and threonine were the main amino acids in ZAV-5. Organic acids and amino acids both contributed weakly to total antioxidant activity of ZAV-5. Catechin, *p*-hydroxybenzoic acid and vanillic acid were the major phenolic compounds in ZAV-5. Phenolic compounds exhibited high antioxidant activity such as caffeic acid, gallic acid, sinapic acid, syringic acid, catechin, rutin, and ferulic acid. Furthermore, amino acids showed synergistic effect on the antioxidant activity of phenolic compounds, while organic acids exhibited antagonistic effect on the antioxidant activity of phenolic compounds. Catechin, vanillic acid and syringic acid had high contribution rates to the antioxidant activity of mixed phenolic compounds.

## Figures and Tables

**Figure 1 molecules-24-03799-f001:**
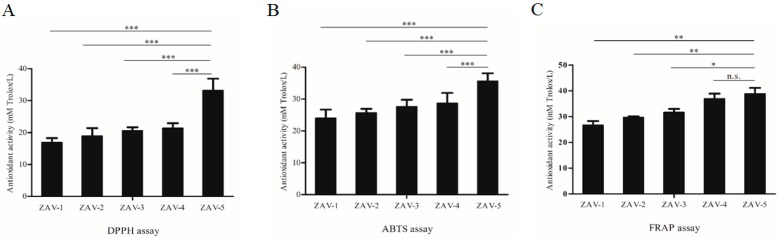
Total antioxidant activity of Zhenjiang aromatic vinegar (ZAV) measured by three different assays. (**A**) DPPH assay, (**B**) ABTS assay, (**C**) FRAP assay. All data are expressed as mean ± S.D. (*n* = 6). * *p* < 0.05, ** *p* < 0.01, *** *p* < 0.001 *vs.* ZAV-5, n.s. indicates no significant difference.

**Figure 2 molecules-24-03799-f002:**
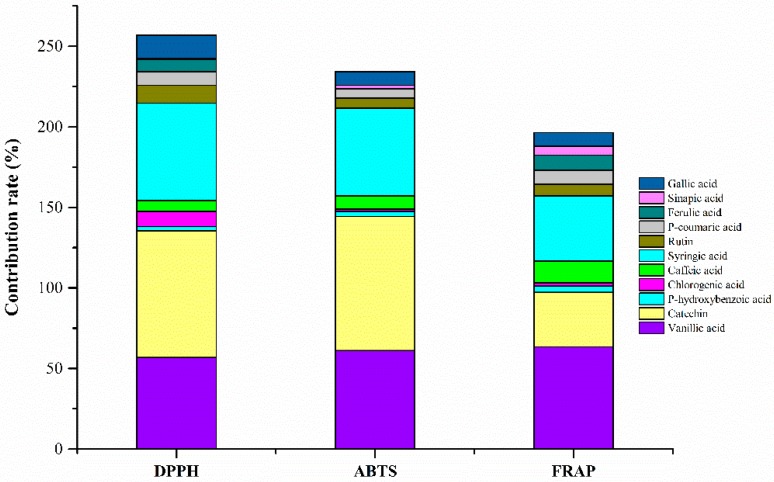
Contribution rate of phenolic compounds in ZAV measured by three different assays.

**Table 1 molecules-24-03799-t001:** Physicochemical parameters of Zhenjiang aromatic vinegar (ZAV) ^1,2^.

Sample Number	pH	Total Acids (g/100mL)	Non-Volatile Acids (g/100mL)	Reducing Sugar (g/100mL)	Soluble Solids (g/100mL)	Amino Nitrogen (g/100mL)
ZAV-1	3.46 ± 0.01 a	4.954 ± 0.221 e	1.769 ± 0.057 e	1.500 ± 0.183 c	12.738 ± 0.084 d	0.399 ± 0.002 c
ZAV-2	3.28 ± 0.02 e	6.192 ± 0.161 d	2.300 ± 0.074 b	1.450 ± 0.071 c	13.616 ± 0.011 c	0.344 ± 0.010 e
ZAV-3	3.34 ± 0.02 d	6.251 ± 0.122 c	2.388 ± 0.089 a	2.050 ± 0.071 b	16.602 ± 0.071 b	0.385 ± 0.007 d
ZAV-4	3.41 ± 0.01 b	6.664 ± 0.078 b	2.079 ± 0.022 c	2.000 ± 0.141 b	16.588 ± 0.022 b	0.427 ± 0.017 b
ZAV-5	3.36 ± 0.02 c	7.726 ± 0.017 a	1.858 ± 0.011 d	2.850 ± 0.212 a	19.428 ± 0.005 a	0.468 ± 0.021 a

^1^ Data are presented as mean *±* S.D. (*n* = 3). ^2^ Significant differences are evaluated using the Duncan Multiple comparison Test. Different letters in the column presents statistically significant differences (*p <* 0.05).

**Table 2 molecules-24-03799-t002:** Organic acid contents and antioxidant activities in ZAV-5 ^1,2,3^.

Organic Acids	Contents (g/100 mL)	Antioxidant Activities (mmol TEAC/g)		
		DPPH	ABTS	FRAP
Oxalic acid	0.188 ± 0.011 f	0.003 ± 0.000 a	ND	0.007 ± 0.000 b
Acetic acid	6.606 ± 0.352 a	0.001 ± 0.000 b	ND	0.002 ± 0.000 d
Tartaric acid	0.091 ± 0.001 g	ND	ND	0.003 ± 0.000 c
Succinic acid	0.529 ± 0.045 d	ND	ND	0.002 ± 0.000 d
Malic acid	0.217 ± 0.016 e	ND	ND	0.003 ± 0.000 c
Pyroglutamic acid	1.363 ± 0.255 c	ND	ND	0.003 ± 0.000 c
Lactic acid	3.283 ± 0.241 b	ND	ND	0.008 ± 0.000 a

^1^ Data are presented as mean *±* S.D. (*n* = 3). ^2^ Significant differences are evaluated using the Duncan Multiple comparison Test. Different letters in the column presents statistically significant differences (*p <* 0.05). ^3^ ND represents not detectable.

**Table 3 molecules-24-03799-t003:** Amino acids contents and antioxidant activities in ZAV-5 ^1,2,3^.

Amino Acids	Contents (g/L)	Antioxidant Activities (mmol TEAC/g)		
		DPPH	ABTS	FRAP
Phenylalanine	0.501 ± 0.033 f	0.127 ± 0.015 a	0.123 ± 0.029 a	0.177 ± 0.015 a
Lysine	0.466 ± 0.024 g	0.012 ± 0.003 b	0.077 ± 0.007 b	0.022 ± 0.003 b
Tyrosine	0.287 ± 0.015 h	0.008 ± 0.001 c	0.015 ± 0.008 c	0.013 ± 0.002 c
Arginine	0.161 ± 0.002 i	ND	0.010 ± 0.002 c	0.006 ± 0.000 d
Valine	0.892 ± 0.041 c	ND	ND	0.005 ± 0.000 e
Isoleucine	0.672 ± 0.051 e	ND	ND	0.006 ± 0.000 d
Leucine	1.535 ± 0.028 a	ND	ND	0.006 ± 0.000 d
Threonine	1.411 ± 0.324 b	ND	ND	0.005 ± 0.000 e
Histidine	0.862 ± 0.079 c	ND	ND	0.006 ± 0.000 d
Alanine	0.775 ± 0.092 d	ND	ND	0.006 ± 0.000 d
Glycine	1.444 ± 0.221 b	ND	ND	0.006 ± 0.001 d
Methionine	0.166 ± 0.011 i	ND	ND	0.006 ± 0.000 d

^1^ Data are presented as mean *±* S.D. (*n* = 3). ^2^ Significant differences are evaluated using the Duncan Multiple comparison Test. Different letters in the column presents statistically significant differences (*p <* 0.05). ^3^ ND represents not detectable.

**Table 4 molecules-24-03799-t004:** Phenolic compounds contents and antioxidant activities in ZAV-5 ^1,2,3^.

Phenolic Compounds	Contents (mg/L)	Antioxidant Activities (mmol TEAC/g)		
		DPPH	ABTS	FRAP
Caffeic acid	2.722 ± 0.014 e	3.323 ± 0.407 a	1.708 ± 0.757 d	2.507 ± 0.156 d
Gallic acid	0.981 ± 0.092 j	3.197 ± 0.186 b	3.174 ± 0.214 a	4.436 ± 0.548 b
Sinapic acid	1.383 ± 0.251 i	2.414 ± 0.478 c	2.134 ± 0.528 c	2.370 ± 0.128 e
Syringic acid	2.123 ± 0.009 f	2.298 ± 0.100 d	2.241 ± 0.721 c	3.285 ± 0.399 c
Catechin	36.745 ± 1.459 a	2.543 ± 0.136 c	2.620 ± 0.393 b	1.266 ± 0.250 f
Rutin	1.812 ± 0.142 g	2.028 ± 0.278 e	2.575 ± 0.082 b	5.094 ± 0.237 a
Ferulic acid	1.575 ± 0.172 h	2.262 ± 0.200 d	2.522 ± 0.700 b	2.440 ± 0.230 e
Chlorogenic acid	6.125 ± 0.912 d	0.291 ± 0.012 f	0.374 ± 0.004 e	0.524 ± 0.001 g
*P*-hydroxybenzoic acid	24.738 ± 2.376 b	0.046 ± 0.004 g	ND	ND
Vanillic acid	15.223 ± 3.892 c	0.029 ± 0.004 h	0.233 ± 0.009 f	0.460 ± 0.078 h
*P*-coumaric acid	1.651 ± 0.224 h	0.027 ± 0.011 h	0.370 ± 0.046 e	0.421 ± 0.003 i

^1^ Data are presented as mean *±* S.D. (*n* = 3). ^2^ Significant differences are evaluated using the Duncan Multiple comparison Test. Different letters in the column presents statistically significant differences (*p <* 0.05). ^3^ ND represents not detectable.

**Table 5 molecules-24-03799-t005:** Effect of main components on antioxidant activities of phenolic compounds in ZAV-5 ^1,2,3^.

Compositions	Antioxidant Activities (mmol TEAC/g)		
	DPPH	ABTS	FRAP
Organic acids	ND	ND	0.038 ± 0.007 f
Amino acids	0.189 ± 0.002 e	0.307 ± 0.020 e	0.364 ± 0.004 e
Phenolic compounds	20.025 ± 2.019 c	21.728 ± 2.985 c	24.462 ± 2.623 c
Organic acids-phenolic compounds	15.075 ± 1.982 d	14.620 ± 1.942 d	20.976 ± 3.755 d
Amino acids-phenolic compounds	35.203 ± 4.361 a	42.371 ± 10.418 a	45.574 ± 9.325 a
Organic acids-amino acids-phenolic compounds	31.979 ± 6.110 b	38.060 ± 7.329 b	39.635 ± 6.927 b

^1^ Data are presented as mean *±* S.D. (*n* = 3). ^2^ Significant differences are evaluated using the Duncan Multiple comparison Test. Different letters in the column presents statistically significant differences (*p <* 0.05). ^3^ ND represents not detectable.

**Table 6 molecules-24-03799-t006:** The samples of ZAV ^1^.

Samples	Number	Aging Time (Year)
ZAV-1	6	0
ZAV-2	6	3
ZAV-3	6	6
ZAV-4	6	8
ZAV-5	6	10

^1^ All the origins are in China, Jiangsu.

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
