# Peer review of "Effects of Organic Acids, Amino Acids and Phenolic Compounds on Antioxidant Characteristic of Zhenjiang Aromatic Vinegar"

_molecules, 2019, doi:10.3390/molecules24203799_

Round 1
Reviewer 1 Report
Comments for Molecules-616341
The study provided references to control the antioxidant characteristic of vinegar through regulating the main compositions, and further improved the quality of vinegar production. I found the paper to be overall well written and much of it to be well described. The design of the field campaign combined with several micro-met stations makes the dataset seem quite useful for the purpose. However, I found some questions in the manuscript. I recommend that a minor revision is warranted. I ask that the authors specifically address each of my comments in their response.
Table1 on page 8 may be changed to supplement 1 , and Table 2 can be changed to Table1 to display Table from the third page. Statistical Analysis in Table1~6 changed Duncan multiple comparison to Duncan’s Multiple comparison Test Figure 1, using three anti-oxidation methods, only showed the ZAV5 has better antioxidant capacity than ZAV1~4. Is ZAV5 has significantly different in each method? There was not shown in Figure1. As for Tables 3 and 6, it was found that the antioxidant concentration of organic acid measured by FRAP was extremely low (0.038 mmol TEAC/g) in ZAV-5. However, the measured antioxidant concentration are relatively lower if compounds were mixed with organic acid. Please discuss the relationship between organic acid and other amino acids or phenolic compounds?Author Response
Response to Reviewer 1 Comments
The study provided references to control the antioxidant characteristic of vinegar through regulating the main compositions, and further improved the quality of vinegar production. I found the paper to be overall well written and much of it to be well described. The design of the field campaign combined with several micro-met stations makes the dataset seem quite useful for the purpose. However, I found some questions in the manuscript. I recommend that a minor revision is warranted. I ask that the authors specifically address each of my comments in their response.
Point 1: Table1 on page 8 may be changed to supplement 1, and Table 2 can be changed to Table1 to display Table from the third page.
Response 1: Thanks for your suggestion. As suggested, we have replaced “Table1” with “Table S1” (Page 8 Line 254-255). “Table 2” has been changed to “Table1” (Page 2 Line 79; Page 3 Line 97). In addition, the orders of the rest tables were changed in our revised manuscript (Page 4 Line 121, 138 and 146; Page 4 Line 149; Page 5 Line 164; Page 6 Line 175, 186 and 197; Page 7 Line 202 and 208).
Point 2: Statistical Analysis in Table1~6 changed Duncan multiple comparison to Duncan’s Multiple comparison Test. Figure 1, using three anti-oxidation methods, only showed the ZAV-5 has better antioxidant capacity than ZAV1~4. Is ZAV-5 has significantly different in each method? There was not shown in Figure1.
Response 2: Thanks for your kindly suggestion. As suggested, we have replaced “Duncan multiple comparison test” with “Duncan’s Multiple comparison Test” in our revised manuscript (Page 3 Line 98-99; Page 4 Line 139-140; Page 5 Line 165-166; Page 6 Line 198-199; Page 7 Line 203-204). In addition, we have added significance analysis between ZAV-5 and other vinegar samples in DPPH、ABTS and FRAP assays (Page 3 Line 114-117). The related images were shown in Figure 1 as follows:
Figure 1. Total antioxidant activity of ZAV measured by three different assays. A: DPPH assay, B: ABTS assay, C: FRAP assay. All data are expressed as mean ± S.D. (n =6). *P < 0.05, **P < 0.01, ***P < 0.001 vs. ZAV-5, n.s. indicates no significant difference.
Figure 1 was replaced in our revised manuscript (Page 3 Line 114). Thanks again for your suggestion to make our manuscript more clearly.
Point 3: As for Tables 3 and 6, it was found that the antioxidant concentration of organic acid measured by FRAP was extremely low (0.038 mmol TEAC/g) in ZAV-5. However, the measured antioxidant concentration are relatively lower if compounds were mixed with organic acid. Please discuss the relationship between organic acid and other amino acids or phenolic compounds?
Response 3: Thanks for your information and suggestion. In the present study, the antioxidant activity of phenolic compounds was significant higher than amino acids and organic acids. When organic acids and phenolic compounds were mixed, antioxidant activities of the mixture were lower than antioxidant activities of phenolic compounds alone. The results suggest that organic acids have antagonistic effect on the antioxidant capacity of phenolic compounds. When organic acids, amino acids and phenolic compounds were mixed, the antioxidant activities of the three main compounds were lower than antioxidant activities of amino acids and phenolic compounds mixture, which also demonstrated the antagonism of organic acids. In addition, amino acids solution showed low antioxidant capacity by three methods. The antioxidant activities of amino acids and phenolic compounds were higher than those of phenolic compounds alone, indicating amino acids had synergistic effect on the antioxidant capacity of phenolic compounds. When organic acids, amino acids and phenolic compounds were mixed, the antioxidant activities of the three main compounds were higher than antioxidant activities of organic acids and phenolic compounds mixture, which also demonstrated the synergism of amino acids. In general, these data suggest that phenolic compounds showed the highest antioxidant activity among the three main compounds. Amino acids showed a synergistic effect on the antioxidant activity of phenolic compounds, while organic acids exhibited an antagonistic effect.
We are sorry for the unclear description. In this study, the antioxidant activity of phenolic compounds was significant higher than amino acids and organic acids. So we mainly discussed the effect of organic acid and amino acids on phenolic compounds. We adjusted the paragraph structure to make our manuscript more clearly and logical.
The related contents were changed in our revised manuscript (Page 7 Line 209-210; Line 213-216; Line 219-222).

Reviewer 2 Report
Manuscript ID: molecules-616341
Title: Effect of Organic Acids, Amino Acids and Phenolic Compounds on Antioxidant Characteristic of Zhenjiang Aromatic Vinegar
General comments
The paper analyses some physicochemical and chemical parameters, and antioxidant activities of Zhenjiang aromatic vinegar.
The basic idea of the manuscript is good, and it could be of practical interest.
However, there are some mistakes and some information is missing
MATERIAL AND METHODS
Which was the number of samples for each aging time?
Why is Table 1 last? In addition, this table shows the pH values, which should go in the results within the physical parameters.
Line 258: Why is Table 2 named here?
Line 260: I think that the method should be better explained. For example, what was the dilution of the samples, how the DPPH solution was prepared, o what method has been followed
Indicate also how the dilution of the samples was made in the rest of methods
Lines 286-290: I think that the method should be better explained: analysis conditions, gradient, standards, etc.
Lines 325-328: Indicate the test (Duncan ???)
RESULTS AND DISCUSSION
List Tables correctly
Lines 96-99. Table 2. Review the statistics
Figure 1. The figures are not seen
I think that the analysis on the results should be improved.
Author Response
Response to Reviewer 2 Comments
The paper analyses some physicochemical and chemical parameters, and antioxidant activities of Zhenjiang aromatic vinegar. The basic idea of the manuscript is good, and it could be of practical interest. However, there are some mistakes and some information is missing.
MATERIAL AND METHODS
Point 1: Which was the number of samples for each aging time?
Response 1: Thanks for your kindly suggestion. 6 vinegar samples for each aging time (0, 3, 6, 8 and 10 years) were obtained from Zhenjiang Hengshun Vinegar Co., Ltd (Jiangsu, China). In addition, the numbers of samples for each aging time were also added in Table S1 as follows:
Table S1. The samples of ZAV 1.
|
Samples |
Number |
Aging time (Year) |
|
ZAV-1 |
6 |
0 |
|
ZAV-2 |
6 |
3 |
|
ZAV-3 |
6 |
6 |
|
ZAV-4 |
6 |
8 |
|
ZAV-5 |
6 |
10 |
1 All the origins are in China, Jiangsu.
The related contents were added in Materials and Methods section of our revised manuscript (Page 8 Line 251 and 253) and Table S1 (Page 8 Line 254-255).
Point 2: Why is Table 1 last? In addition, this table shows the pH values, which should go in the results within the physical parameters.
Response 2: Thanks you for your kindly reminding. Table1 has been changed to Table S1 (Page 8 Line 254-255). In addition, the pH values were added in Table 1 Physicochemical parameters of ZAV (Page 3 Line 97) as follows:
Table 1. Physicochemical parameters of ZAV 1,2.
|
Sample number |
pH |
Total acids (g/100mL) |
Non-volatile acids (g/100mL) |
Reducing sugar (g/100mL) |
Soluble solids (g/100mL) |
Amino nitrogen (g/100mL) |
|
ZAV-1 |
3.46 ± 0.01 a |
4.954 ± 0.221 e |
1.769 ± 0.057 e |
1.500 ± 0.183 c |
12.738± 0.084 d |
0.399 ± 0.002 c |
|
ZAV-2 |
3.28 ± 0.02 e |
6.192 ± 0.161 d |
2.300 ± 0.074 b |
1.450 ± 0.071 c |
13.616± 0.011 c |
0.344 ± 0.010 e |
|
ZAV-3 |
3.34 ± 0.02 d |
6.251 ± 0.122 c |
2.388 ± 0.089 a |
2.050 ± 0.071 b |
16.602± 0.071 b |
0.385 ± 0.007 d |
|
ZAV-4 |
3.41 ± 0.01 b |
6.664 ± 0.078 b |
2.079 ± 0.022 c |
2.000 ± 0.141 b |
16.588± 0.022 b |
0.427 ± 0.017 b |
|
ZAV-5 |
3.36 ± 0.02 c |
7.726 ± 0.017 a |
1.858 ± 0.011 d |
2.850 ± 0.212 a |
19.428± 0.005 a |
0.468 ± 0.021 a |
1 Data are presented as mean ± S.D. (n = 3). 2 Significant differences are evaluated using the Duncan Multiple comparison Test. Different letters in the column presents statistically significant differences (P < 0.05).
Point 3: Line 258: Why is Table 2 named here?
Response 3: Thanks for your kindly reminding. The description of Table 2 should be showed in the results. We have deleted the sentence “General components of the vinegar samples were presented in Table 2” in our revised manuscript (Page 9 Line 262).
Point 4: Line 260: I think that the method should be better explained. For example, what was the dilution of the samples, how the DPPH solution was prepared, o what method has been followed. Indicate also how the dilution of the samples was made in the rest of methods.
Response 4: Thanks for your suggestion. As suggested, we have explained DPPH, ABTS and FRAP assays in more details. The description of the methods was added as follows:
DPPH assay. Every sample was diluted 20 times with distilled water. DPPH working solution was prepared. 20 mg of DPPH was added to 250 mL ethanol to make 0.2 mmol/L DPPH solution. 0.2 mmol/L DPPH solution and ethanol (5:4, v/v) were mixed to make DPPH working solution. All the solutions were incubated at room temperature in the dark. Diluted samples (20 μL) were added to DPPH working solution (180 μL), and the mixed solution were incubated at room temperature for 30 min in the dark. The absorbances of solution were measured at 517 nm using an automated enzyme-linked immunosorbent assay (ELISA) reader (Tecan Austria GmbH, Salzburg, Austria). The data were expressed as trolox equivalent antioxidant capacity (TEAC).
ABTS assay. Every sample was diluted 20 times with distilled water.
FRAP assay. Every sample was diluted 20 times with distilled water.
The related contents were added in Materials and Methods section of our revised manuscript (Page 9 Line 264-267, 273 and 280).
Point 5: Lines 286-290: I think that the method should be better explained: analysis conditions, gradient, standards, etc.
Response 5: Thanks for your kindly suggestion. As suggested, we have added the information in 3.5. Detection of Amino Acids. The description of the methods was revised as follows:
Amino acid contents in vinegar samples were analyzed by Amino Acid Analyzer (S-433D; SYKAM, Munich, Germany) equipped with a fluorescence detector. Vinegar samples were centrifuged at 7000 rpm for 15 min at a room temperature, diluted 50 times with distilled water. 200 µL of diluted vinegar samples and 800 µL of salicylic acid (2 %, v/v) were mixed and stood at room temperature for 30 min. The mixture was centrifuged at 13000 rpm for 15 min, and the upper layer was filtered via 0.45 µm Millipore membrane. LCAK06/Na (4.6 mm × 150 mm) column was used for free amino acid determination. The working parameters were 0.45 mL/min flow rate, 50 μL injection volume and 58 °C column temperature. The ninhydrin (NIN)-derivatized amino acids were determinated at 570 nm and at 440 nm. A calibration curve was prepared with different concentrations of amino acids solutions, and the results were expressed as g/L of vinegar samples.
The related contents were added in Materials and Methods section (Page 9 Line 297-298 and Line 301-305).
Point 6: Lines 325-328: Indicate the test (Duncan ???)
Response 6: We are sorry for the unclear description. Duncan multiple comparison test refers to the hypothesis test of whether there is significant difference between the mean of each sample. Variance analysis was carried by duncan multiple comparison test using the SPSS statistical package version 24.0 software.
The related contents were added in Materials and Methods section 3.10. Statistical Analysis (Page 10 Line 341-343).
RESULTS AND DISCUSSION
Point 7: List Tables correctly
Response 7: Thanks for your kindly suggestion. As suggested, table lists have been checked and corrected (Page 3 Line 97; Page 4 Line 121, 138 and 146; Page 4 Line 149; Page 5 Line 164; Page 6 Line 175, 186 and 197; Page 7 Line 202 and 208).
Point 8: Lines 96-99. Table 2. Review the statistics
Response 8: Thanks for your kindly suggestion. We have been checked carefully all the data in Table1 (Page 3 Line 97).
Point 9: Figure 1. The figures are not seen. I think that the analysis on the results should be improved.
Response 9: Thanks for your kindly suggestion. As suggested, we have improved the quality of Figure 1. The related images were shown in Figure 1 as follows:
Figure 1. Total antioxidant activity of ZAV measured by three different assays. A: DPPH assay, B: ABTS assay, C: FRAP assay. All data are expressed as mean ± S.D. (n =6). *P < 0.05, **P < 0.01, ***P < 0.001 vs. ZAV-5, n.s. indicates no significant difference.
Figure 1 was replaced in our revised manuscript (Page 3 Line 114). Thanks again for your suggestion to make our manuscript more clearly.
Reviewer 3 Report
Manuscript is well written with appropriate references. The authors have done a god job presenting and discussing the data. I recommend this manuscript for publication.
Author Response
Comments and Suggestions for Authors
Manuscript is well written with appropriate references. The authors have done a good job presenting and discussing the data. I recommend this manuscript for publication.
Round 2
Reviewer 2 Report
General comments
The manuscript has been revised and it is much improved